molecular biology/genomics/plant science

rice, chilling stress, booting stage, RNA-seq, differentially expressed genes

**Authors for correspondence:**
Guojun Pan
e-mail: panguojun777@163.com
Tao Guo
e-mail: guo.tao@vip.163.com

[†]Zhenhua Guo and Lijun Cai contributed equally to this study.
[‡]Guojun Pan and Tao Guo contributed equally to this study.

# Global analysis of differentially expressed genes between two Japonica rice varieties induced by low temperature during the booting stage by RNA-Seq

Zhenhua Guo[1,2,†], Lijun Cai[3,†], Chuanxue Liu[1], Cuihong Huang[2], Zhiqiang Chen[1], Guojun Pan[1,‡] and Tao Guo[2,‡]

[1]Rice Research Institute of Heilongjiang Academy of Agricultural Sciences, Jiamusi, Heilongjiang, People's Republic of China
[2]National Engineering Research Center of Plant Space Breeding, South China Agricultural University, Guangzhou, Guangdong, People's Republic of China
[3]Jiamusi Branch of Heilongjiang Academy of Agricultural Sciences, Jiamusi, Heilongjiang, People's Republic of China

ZG, 0000-0002-6745-2322; LC, 0000-0002-9003-6191; GP, 0000-0002-6067-5363; TG, 0000-0002-2707-1760

As one of the abiotic stresses, low temperature severely threatens rice production during its entire growth period, especially during the booting stage. In the present study, transcriptome analysis was performed comparing Longjing (LJ) 25 (chilling-tolerant) and LJ 11 (chilling-sensitive) rice varieties to identify genes associated with chilling tolerance in rice spikelets. A total of 23 845 expressed genes and 13 205 differentially expressed genes (DEGs) were identified, respectively. Gene ontology (GO) enrichment analyses revealed 'response to cold' (containing 180 DEGs) as the only category enriched in both varieties during the entire cold treatment period. Through MAPMAN analysis, we identified nine and six DEGs related to the Calvin cycle and antioxidant enzymes, respectively, including *OsRBCS3*, *OsRBCS2*, *OsRBCS4*, *OsAPX2* and *OsCATC*, that under chilling stress were markedly downregulated in LJ11 compared with LJ25. Furthermore, we predicted their protein–protein interaction (PPI) network and identified nine hub genes (the threshold of co-expressed gene number ≥ 11) in Cytoscape, including three RuBisCO-related genes with 14 co-expressed genes. Under chilling stress, antioxidant enzyme activities

(peroxidase (POD) and catalase (CAT)) were downregulated in LJ11 compared with LJ25. However, the content of malondialdehyde (MDA) was higher in LJ11 compared with LJ25. Collectively, our findings identify low temperature responsive genes that can be effectively used as candidate genes for molecular breeding programmes to increase the chilling tolerance of rice.

## 1. Introduction

Growing plants are highly sensitive to and unable to escape from environmental stress. Abiotic stresses include drought, high salinity, high alkalinity, flooding and high or low temperature. Low-temperature stress is usually defined as freezing injury (less than 0°C) or chilling injury (0–15°C) according to the plant's responses [1]. Rice, as a model plant for research and one of the most important cereal crops feeding more than half of the world's population, originates from tropical and temperate climate zones and is sensitive to low temperatures [2]. Among two major varieties of Asian cultivated rice (*Oryza sativa*), the chilling tolerance of *japonica* rice (*O. sativa* ssp. *japonica*) is stronger than that of *indica* rice (*O. sativa* ssp. *indica*) [3].

Rice is threatened by low temperatures during its entire growing period, showing different responses to chilling stress during each growth stage, such as a low seed germination rate, chlorotic leaves, reduced culm length and reduced numbers of tillers during the germination stage and the vegetable stage. However, the booting stage is the most sensitive to low temperatures, and the chilling injury occurring during this stage is called obstructive chilling damage, which leads to pollen sterility and subsequent yield losses. Previous research has shown that the development of tapetum cells from the tetrad to the young microspore stage (YM stage) was crucial to pollen maturation for it is the period most affected by low temperature. *CYP704B2* [4], *OsUgp1* [5], *OsUgp2* [6], *OsMTR1* [7], *Osg1* [8], *OsC6* [9] and *OSINV4* [10] are all functionally associated with tapetum development and, if disturbed by chilling stress, the resulting tapetum abnormity and pollen abortion will subsequently lead to a decline in yield. As the degradation of the tapetum is indispensable to microspore development and pollen maturation, the punctuality of the normal programmed cell death (PCD) of the tapetum cells is essential, which, if advanced or delayed, would lead to pollen sterility. The tapetum is a model organ for plant PCD investigation, and several regulating genes associated with tapetum degradation have been reported, including *TDR* [11], *DTC* [12], *OsMS1* [13], *API5* [14], *EAT1* [15], *OsADF* [11] and *OsMYB80* [16].

In addition to the effect of low temperature on morphological and histological traits, there are physiological and biochemical metabolites, such as soluble sugars, amino acids and phytohormones that are stress-related and that are useful indicators of chilling stress. Soluble sugars, proline and antioxidants are osmolytes and cryoprotectants accumulating at low temperatures to alleviate osmoregulation and reduce oxidative damage [17]. Among the phytohormones, abscisic acid (ABA) is indispensable for adaptive responses to various abiotic stresses [18]. Stress adaption can be divided into ABA-dependent and ABA-independent pathways [19]. In rice, the ABA-responsive genes (NAC recognition sequence (NACRS)-containing target gene) will be activated by ABA accumulation during chilling stress through *cis*-acting ABA-response elements (ABRE) (NACRS-containing target gene) and the ABRE-binding bZIP transcription factor (ABF) leading to increased cold tolerance [20]. The dehydration-responsive element-binding proteins-C repeat/dehydration-responsive elements (DREB-CRT/DRE) pathway activated by $Ca^{2+}$ signalling cascade [21], and the mitogen-activated protein kinase (MAPK) signal transduction networks (MAPKK-MAPK) (*OsMKK6-OsMPK3*) pathway induced by the accumulation of reactive oxygen species (ROS) [22], are the two ABA-independent pathways in rice involved in the adaptation to low temperature. Therefore, chilling tolerance of rice is a very complex quantitative trait controlled by multiple loci as well as by the environment.

With the recent development of the high-throughput sequencing technology, which has been applied in various plant species, plenty of remarkable achievements have been obtained. RNA sequencing (RNA-Seq) provides a far more accurate way to identify transcriptional profiling and differentially expressed genes (DEGs) than other high-throughput approaches [23]. Moreover, as a model plant of gramineous monocots, the rice genome has been completely sequenced and could be used as the reference genome for the de novo generation of transcriptomes. Here, combined with de novo transcriptome analysis, we aimed to identify candidate genes capable of regulating the chilling tolerance of rice at the booting stage in cold regions, which could be significant for determining the mechanism of chilling tolerance and reducing the yield losses caused by low temperature in cold regions. In the present study, RNA-Seq was performed to analyse the DEGs at three different points of low temperature treatment in a novel chilling-tolerant cultivar Longjing 25 (LJ25). The aim was to

comprehensively understand the dynamic changes in the transcriptomes of the chilling-tolerant rice at low temperature and the characteristics of chilling-induced transcriptomes.

# 2. Material and methods

## 2.1. Plant materials and low temperature treatment

The rice variety (*Oryza sativa* L. ssp. *japonica*) LJ25, is known for its chilling tolerance at the booting stage (6.7–8.1% spikelet sterility) [24], while Longjing 11 (LJ11) is a chilling-sensitive cultivar with around 94% spikelet sterility at the booting stage under the same chilling stress at the booting stage [25]. Plants were subjected to the cool air treatment method [26]. For both LJ25 and LJ11, 20 plants were sown in a circle in plastic pots (diameter 25 cm and height 23 cm) and grown in a phytotron (28°C/22°C day/night, 80% RH, 12 h-light/12 h-dark photoperiod) after germination. In order to keep the growth process synchronously, only the main culm of each plant was left. The auricle distance (AD) method was used to identify the period of pollen development [27]. Since the meiosis during the booting stage was the most sensitive stage to chilling stress, the pollen was thought to undergo this stage when the auricle of the flag leaf was approximately 5 cm below the auricle of the penultimate leaf. At this stage, half of the plants were transferred to another phytotron and maintained at 12°C under otherwise identical conditions for the specified time (2 or 4 days). After this, the cold-treated plants were moved back to the original phytotron until they reached maturity. Chilling tolerance was evaluated on the basis of the mean percentage of empty grains of the main spikelet.

## 2.2. Sample preparation and RNA extraction

Fresh young spikelets (0.5 g) (3.5–4.5 mm in length) were removed from the upper third of the panicles and collected at day 0, day 2 and day 4 during the incubation at 12°C. Spikelets collected on day 0 were regarded as controls. Samples were immediately frozen in liquid $N_2$ and stored at −80°C until analysed, each sample with three replicates. Total RNA was extracted using a total RNA extraction kit (Invitrogen, Carlsbad, CA, USA) according to the manufacturer's instructions. The quantity and quality of the total RNA were confirmed with a NanoDrop™ 2000 (Thermo Scientific™, Waltham, MA, United States). The purified RNA was used for library construction and RNA-sequencing.

## 2.3. Determination of physiological traits under chilling stress

Fresh young spikelets were collected at day 0, day 2 and day 4 of the chilling stress treatment as described above for physiological determination. With a precooled mortar and pestle, about 50 mg of frozen spikelets from each experimental group was milled with 50 mM potassium phosphate buffer (pH 7.8) containing 1% polyvinylpyrrolidone. After homogenization and centrifugation at $15\,000g$ for 20 min at 4°C, the supernatant contained the crude enzyme preparation. Superoxide dismutase (SOD) activity was measured with the photochemical method of nitroblue tetrazolium (NBT) as described by Giannopolitis & Ries [28] with minor modifications. Briefly, 1.5 ml phosphate buffer solution (0.05 mol $l^{-1}$), 0.3 ml methionine (130 mmol $l^{-1}$), 0.3 ml NBT (750 µmol $l^{-1}$), 0.3 ml EDTA-2Na (100 µmol $l^{-1}$), 0.3 ml riboflavin solution (20 µmol $l^{-1}$) and 0.25 ml distilled water were added to 0.05 ml crude enzyme extract. The absorbance of the mixture was measured at 560 nm, and one unit of SOD activity was defined as the amount of enzyme required to inhibit 50% of NBT.

Peroxidase (POD) activity was determined using previously reported protocols [29]. A substrate mixture containing 0.9 ml guaiacol solution (0.2%), 1 ml $H_2O_2$ (30%) and 1 ml PBS (0.05 mol $l^{-1}$, pH 6.0) were added to the enzyme extract (1 ml). The activity of POD was measured based on the change of absorbance of the brown guaiacol substrate at 460 nm. Catalase (CAT) activity was assayed following the method reported by Miao *et al*. [30] and with minor modifications. The reaction mixture contained 3 ml PBS (0.1 mol $l^{-1}$, pH 7.8), 0.6 ml $H_2O_2$ (30%) and 0.1 ml enzyme extract. CAT activity was calculated as the decrease of the absorbance at 240 nm for 1 min following the decomposition of $H_2O_2$. Malondialdehyde (MDA) content was determined using the method reported by Yang *et al*. [31] with minor changes. Briefly, 0.3 g rice spikelets (3.5–4.5 mm length) were collected and homogenized in 5 ml trichloroacetic acid (10%) which contained 0.25% thiobarbituric acid. The mixture was incubated for 30 min at 95°C and then cooled in ice water and centrifuged at $10\,000g$ for 20 min. The absorbance of the supernatant was measured at 440, 532 and 600 nm to determine MDA content. All data were analysed by the SAS statistics program, and significant statistical differences are set as $p < 0.05$.

## 2.4. RNA-sequencing and data analysis

For the construction of sequencing libraries, mRNA was purified twice with oligo-(dT) attached magnetic beads and fragmented with an RNA fragmentation kit according to the manufacturer's protocol (Illumina, San Diego, CA, USA). Random primers and reverse transcriptase were used to synthesize the first-strand cDNA by using the mRNA fragments as templates, followed by second-strand cDNA generation and adaptor ligation. After isolating through gel electrophoresis and PCR amplification (18 cycles), the products of the cDNA fragments (200 bp) were uploaded onto an Illumina Hiseq™ 2500 platform and subjected to paired-end ($2 \times 150$ bp) sequencing according to the manufacturer's protocols at Beijing Biomarker Biotechnology Co. (Beijing, China). The raw reads were generated first and appeared in FASTQ format, including adapter-containing reads, low-quality reads with $Q$-value $\leq 10$ or unknown nucleotides (greater than 10%). After filtering, the clean reads were assembled using the short-read assembly program Trinity [32] and clustered using the TIGR gene indices clustering (TGICL) tools into unigenes [33]. The genome of *Oryza sativa* L. ssp. *japonica* (Oryza_sativa_IRGSP-1.0) was set as a reference. The unigene expression abundance was defined as the number of fragments uniquely matching a particular rice gene and normalized to fragments per kilobase of transcript per million fragments mapped (FPKM). BLAST alignment was performed between unigenes and protein databases with the BLASTx program (https://blast.ncbi.nlm.nih.gov) at an E-value threshold of $1 \times 10^{-5}$, including the NCBI non-redundant (NR) database (http://www.ncbi.nlm.nih.gov), and the Swiss-Prot protein database (http://www.expasy.ch/sprot). The edgeR package (1.10.1) [34] was used to identify DEGs across samples. Genes were considered as significant DEGs with the selection criteria of |fold change| (|FC|) $\geq 3$ and a false discovery rate (FDR) $< 0.01$, and then subjected to gene ontology function enrichment (GO; http://www.geneontology.org) and Kyoto Encyclopedia of Genes and Genomes (KEGG; http://www.genome.jp/kegg) pathway analyses. R package heatmap.3 was used to performed heatmaps [35]. All the analysis above was performed with the related tools on the online open platform BMKCloud (http://www.biocloud.net/). MᴀᴘMᴀɴ software v. 3.5.1R2 [36] was applied to integrate the significant genes into diverse overviews. The protein–protein interactions (PPI) of the DEGs were predicted with the STRING database [37] of Rice PPI data and visualized in Cytoscape [38]. A combined score greater than 0.5 was set as the cut-off for the predicted PPI pairs.

## 2.5. Validation of DEGs by RT-PCR

Ten DEGs (electronic supplementary material, table S1) were selected for validation by quantitative reverse transcription-polymerase chain reaction (qRT-PCR). Total RNA of each sample was extracted as described above, treated with DNase, followed by first-strand cDNA synthesis using Super-Script II RNase H−Reverse Transcriptase (Takara, Japan). SYBR-based qRT-PCR reactions (SYBR Green I, Osaka, Japan) were performed on an ABI StepOne Plus system. The qRT-PCR results were evaluated by the $2^{-\Delta\Delta CT}$ method [38]. Each candidate gene was detected in triplicate and averaged as mean value ± standard error. The reference gene used was *Actin1*.

# 3. Results

## 3.1. Effect of chilling stress at the booting stage on the number of empty grains of the two rice varieties

The percentage of empty grains of the two rice varieties LJ11 and LJ25 was determined after 0, 2 and 4 days of chilling stress (12°C) (figure 1). As shown, the percentage of empty grains in LJ11 significantly increased after 2 days of chilling stress, and again increased significantly between day 2 and day 4 of chilling stress. After 4 days of chilling treatment, more than 90% of the grains of strain LJ11 were empty. By contrast, for LJ25, there was nearly no change in the number of empty grains after 2 days of treatment, a significant increase only occurring after 4 days chilling stress.

## 3.2. Assembly of RNA-Seq reads and data analysis

In total, 18 libraries were constructed, including triplicates from two varieties with three chilling stress time points each. 1055.7 Mb clean reads were generated. An average of 81.67% of the sequencing reads was mapped to the reference genome (Oryza_sativa_IRGSP-1.0), while the ratio of the multiple mapped reads was 3.3–9.96%. To guarantee the accuracy of follow-up research, the multiple mapped reads

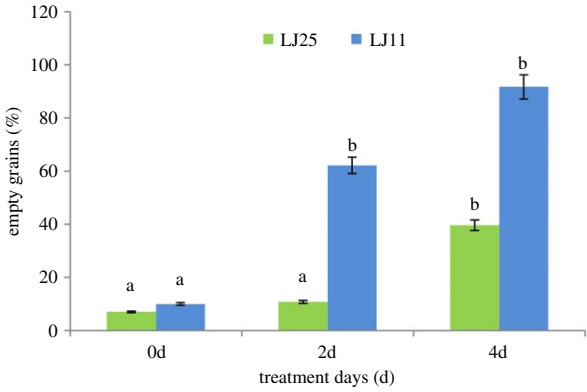

**Figure 1.** Effect of chilling stress on the percentage of empty grains in LJ25 and LJ11 under different chilling stress periods. The letters represent significant differences of comparison among control group (0d) and treatment groups (2d and 4d) for each genotype. The letter 'a' represents no significant difference of comparison between two genotypes at each treatment, the letter 'b' represents significant differences of comparison between two genotypes at each treatment with $p < 0.05$.

**Table 1.** Summary of the quality of sequencing data and the statistics of the transcriptome assembly.

| samples | clean reads | GC content | % ≥ Q30 | mapped reads | multiple map reads |
|---------|-------------|------------|---------|--------------|--------------------|
| 25-0-1 | 43 219 584 | 55.53% | 92.43% | 36 434 612 (84.30%) | 4 273 029 (9.89%) |
| 25-0-2 | 57 834 352 | 56.07% | 91.78% | 47 992 825 (82.98%) | 2 727 536 (4.72%) |
| 25-0-3 | 61 653 378 | 55.56% | 91.98% | 51 148 445 (82.96%) | 3 291 772 (5.34%) |
| 25-2-1 | 55 764 392 | 55.52% | 91.46% | 45 602 317 (81.78%) | 2 931 865 (5.26%) |
| 25-2-2 | 55 515 810 | 54.52% | 91.74% | 45 293 095 (81.59%) | 2 996 223 (5.40%) |
| 25-2-3 | 49 870 338 | 55.39% | 92.05% | 41 000 921 (82.22%) | 3 024 240 (6.06%) |
| 25-4-1 | 60 499 320 | 55.28% | 91.94% | 49 312 142 (81.51%) | 3 266 350 (5.40%) |
| 25-4-2 | 61 702 412 | 55.05% | 91.96% | 51 115 346 (82.84%) | 4 546 560 (7.37%) |
| 25-4-3 | 69 395 674 | 55.67% | 91.35% | 56 124 178 (80.88%) | 2 601 713 (3.75%) |
| 11-0-1 | 53 942 024 | 56.58% | 91.10% | 43 361 427 (80.39%) | 1 779 094 (3.30%) |
| 11-0-2 | 68 461 348 | 56.40% | 91.52% | 55 243 173 (80.69%) | 3 417 620 (4.99%) |
| 11-0-3 | 62 525 964 | 55.92% | 91.26% | 50 368 723 (80.56%) | 4 155 318 (6.65%) |
| 11-2-1 | 43 221 480 | 54.77% | 91.39% | 34 915 829 (80.78%) | 2 243 227 (5.19%) |
| 11-2-2 | 74 234 502 | 55.35% | 91.86% | 60 514 688 (81.52%) | 2 975 696 (4.01%) |
| 11-2-3 | 62 994 916 | 55.02% | 91.29% | 50 799 160 (80.64%) | 3 521 217 (5.59%) |
| 11-4-1 | 61 814 654 | 53.92% | 91.86% | 49 175 101 (79.55%) | 6 156 701 (9.96%) |
| 11-4-2 | 59 057 624 | 53.98% | 91.92% | 48 432 689 (82.01%) | 5 772 643 (9.77%) |
| 11-4-3 | 54 019 710 | 53.40% | 92.20% | 44 750 148 (82.84%) | 4 243 560 (7.86%) |

should be excluded since they would reduce the accuracy and coverage degree in transcriptomic studies using the short-read sequencing methodology. In addition, the GC contents ranged from 53.4% to 56.58%, while the quality score (Q30) was 91.1–92.45% (table 1). The FPKM method was used to calculate gene expression level, and those with less than five FPKMs were regarded as low expressed genes and filtered out. In total, 23 845 expressed genes were generated. In LJ25, 22 516, 22 258 and 21 607 genes were identified after 0, 2 and 4d days chilling treatment, respectively, while there were 22 972, 21 269 and 21 004 genes in LJ11, respectively (figure 2; electronic supplementary material, table S2).

## 3.3. Identification and functional annotation of DEGs under chilling stress

Using │FC│ ≥ 3 and FDR < 0.01 as the selection criteria of significant DEGs in this study, a total of 13 205 DEGs were identified, of which 9030 (4672 up- and 4358 downregulated), 8160 (4762 up- and 3398

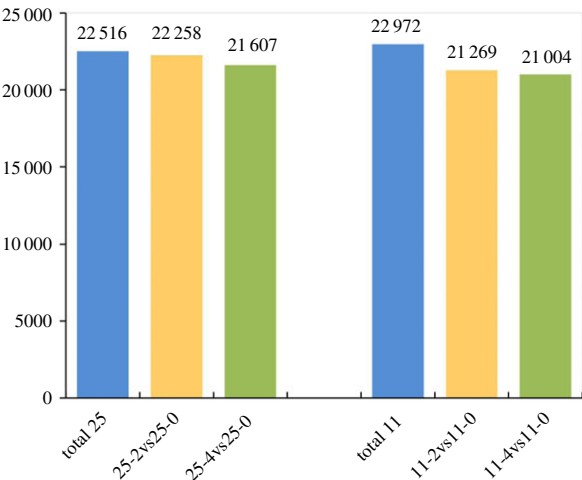

**Figure 2.** Histogram of expressed genes of LJ11 and LJ25 under different chilling stress periods. The numbers above the columns represent the number of expressed genes identified after 0, 2 and 4 days of chilling stress on LJ25 and LJ11, respectively.

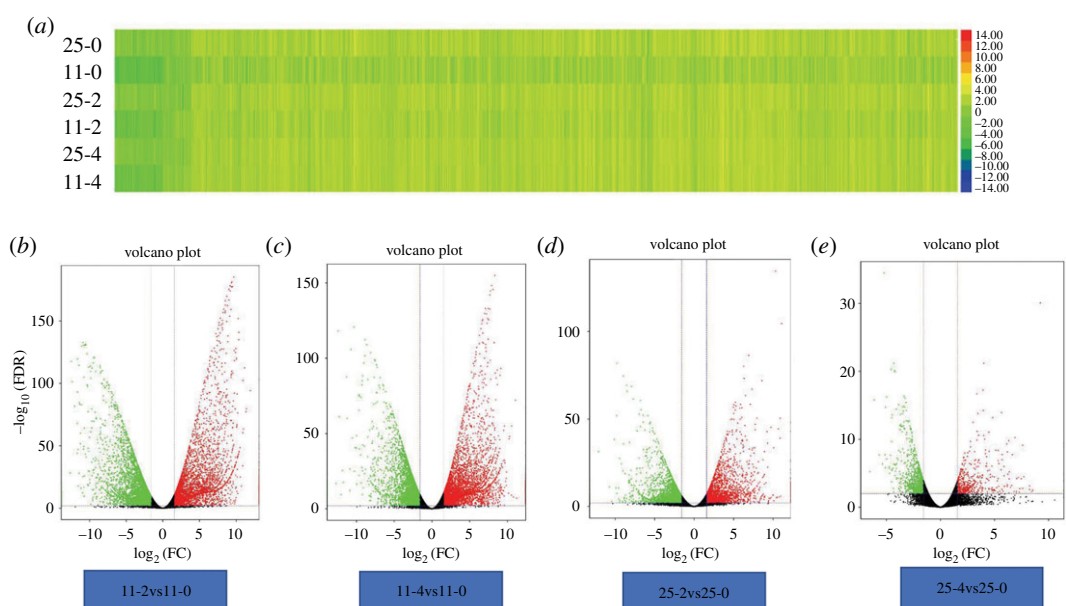

**Figure 3.** Statistical analysis of DEGs in LJ25 and LJ11 during chilling stress. (*a*) Heatmap displaying the expression levels of all identified DEGs in both LJ25 and LJ11 after 0, 2 and 4 days of chilling stress. (*b*–*e*) Volcano Plots of FDR versus fold change displaying the DEGs identified in compared groups: 11-2vs11-0, 11-4vs11-0, 25-2vs25-0 and 25-4vs25-0, respectively, as indicated. The green dots represent downregulated genes, the red dots represent upregulated genes and the black dots represent genes that are not differentially expressed.

downregulated), 4038 (2124 up- and 1914 downregulated) and 813 (258 up- and 555 downregulated) were identified in 11–2vs11-0 (LJ11 day 2 versus day 0), 11-4vs11-0, 25-2vs25-0, and 25-4vs25-0, respectively (figure 3*a,b*; electronic supplementary material, table S3). As shown in each comparison, there were obviously more DEGs in LJ11 than in LJ25, and there was no obvious difference between the number of DEGs in LJ11 after 2 and 4 days of chilling stress. However, most of the genes were differentially expressed after 2 days of chilling treatment in LJ25 and then returned to a normal expression level after 4 days of chilling treatment. These results showed that the low temperature affected the gene expression levels more strongly and continuously in LJ11 than in LJ25, indicating that the tolerance and adaptability of LJ25 to low temperature was stronger than that of LJ11.

To ascertain the biological functions of the DEGs in the four compared groups (LJ25 day 0 versus day 2, LJ25 day 0 versus day 4, LJ11 day 0 versus day 2, and LJ11 day 0 versus day 4), we studied the GO terms

within the 'biological process' (BP) category. Taking the correct $p$-value Kolmogorov–Smirnov (KS) < 0.005 as the threshold, 110 BP GO terms were identified in LJ11 after both 2 and 4 days of chilling treatment, while 57 and 72 BP GO terms were detected in LJ25 after 2 and 4 days of chilling treatment, respectively (electronic supplementary material, table S4). The top 20 BP GO terms with the highest percentage of significant genes (% = significant/annotated) within each compared group are shown in figure 4. There were nearly 60 BP GO terms altogether. The top BP GO term in LJ11 after 2 and 4 days of chilling stress was 'photosynthetic electron transport in photosynthesis' (GO: 0009773) and 'photosynthesis, light harvesting' (GO: 0009765), in which 79.07% and 86.21% of the annotated genes were significant, respectively. On the other hand, in LJ25 'monosaccharide transport' (GO: 0015749) was the top BP GO category with the highest percentage of significant genes (52.17%) after 2 days of chilling stress, and 'callose deposition in phloem sieve plate' (GO: 0080165) was the one with the highest percentage (71.43%) after 4 days of chilling stress. In addition, the percentage of significant genes was generally higher in LJ11 than in LJ25, and most of them were assigned to the categories related to photosynthesis and energy metabolisms. However, the category 'response to cold' (GO: 0009409) was the only category enriched in all four groups (four colours in figure 4; electronic supplementary material, table S4), and therefore we subsequently focused on this GO category.

## 3.4. MapMan analysis of the DEGs in GO category: 'response to cold' in LJ25 and LJ11 under chilling stress at the booting stage

A total of 180 DEGs in the GO category: 'response to cold' were identified from all four compared groups and confirmed by searching the Biocloud platform (http://www.biocloud.net/) (figure 5a; electronic supplementary material, table S6). As the MapMan program is a very effective tool for visualizing diverse results associated with high-throughput RNA-Seq data [39], we used it to examine those DEGs. After uploading the DEGs to the MapMan program, 21 DEGs were located in the 'protein' category, 20 in 'miscellaneous function' (misc), 14 in 'photosynthesis' (PS), 13 in 'lipid metabolism', 10 in 'stress', six in 'redox', eight in 'RNA', six in 'signalling' and a smaller number in other categories (figure 6a; electronic supplementary material, table S6–1). However, 33 genes were not located in any MapMan terms. Furthermore, we analysed the metabolism overview using these 180 DEGs, and according to the 49 DEGs found here, 14 belonged to the 'PS' and 13 to the 'lipid metabolism' as described above; three each to 'minor carbohydrate (CHO) metabolism' and 'major CHO metabolism'; two each to 'cell wall', 'amino acid metabolism', 'tricyclic acid cycle' (TCA), 'nitrogen-metabolism', 'nucleotide metabolism' and 'oxidative phosphorylation' (OPP), and just one gene each to 'secondary metabolism', 'gluconeogenesis' and 'redox' (figure 6b; electronic supplementary material, table S6–1).

## 3.5. Analysis of photosynthesis overview on DEGs in GO category: 'response to cold'

As photosynthesis is crucial to plant growth and development, we investigated the 14 DEGs contained in this pathway (figure 6c; electronic supplementary material, table S6–2). As shown in figure 6c, nine genes were assigned to Calvin cycle, encoding fructose-1–6-bisphosphatase (Os03g0267300), glyceraldehyde 3-phosphate dehydrogenase (Os03g0129300), phosphoglycerate kinase (Os05g0496200), phosphoribulokinase (Os02g0698000), ribulose-phosphate 3-epimerase (Os03g0169100) and ribulose 1,5 bisphosphate carboxylase/oxygenase (RuBisCO) (Os12g0291400, Os12g0274700, Os12g0292400), respectively. Four genes were assigned to the light reaction, encoding ATP synthase subunits (Os10g0355800, Os03g0278900), the cytochrome b6-f complex iron-sulfur subunit (Os07g0556200) and the oxidoreductase NAD-binding domain (Os06g0107700), while one gene (Os03g0738400) was assigned to photorespiration, encoding serine hydroxymethyltransferase. Moreover, these genes were all downregulated in LJ11 after 2 and 4 days of chilling stress. Especially the three genes encoding the small subunit of RuBisCO, OsRBCS3, OsRBCS2 and OsRBCS4, were sharply downregulated after chilling stress (figure 5b–e).

## 3.6. Analysis of biotic stress overview with elements involved in the signalling pathways on DEGs in GO category: 'response to cold'

To survey the significant biotic stress pathway of elements involved in the signalling pathways involved in response to chilling stress, we analysed the MapMan biotic stress overview associated with those DEGs (figure 6d). A total of 48 DEGs were divided into seven categories in the pathway (electronic

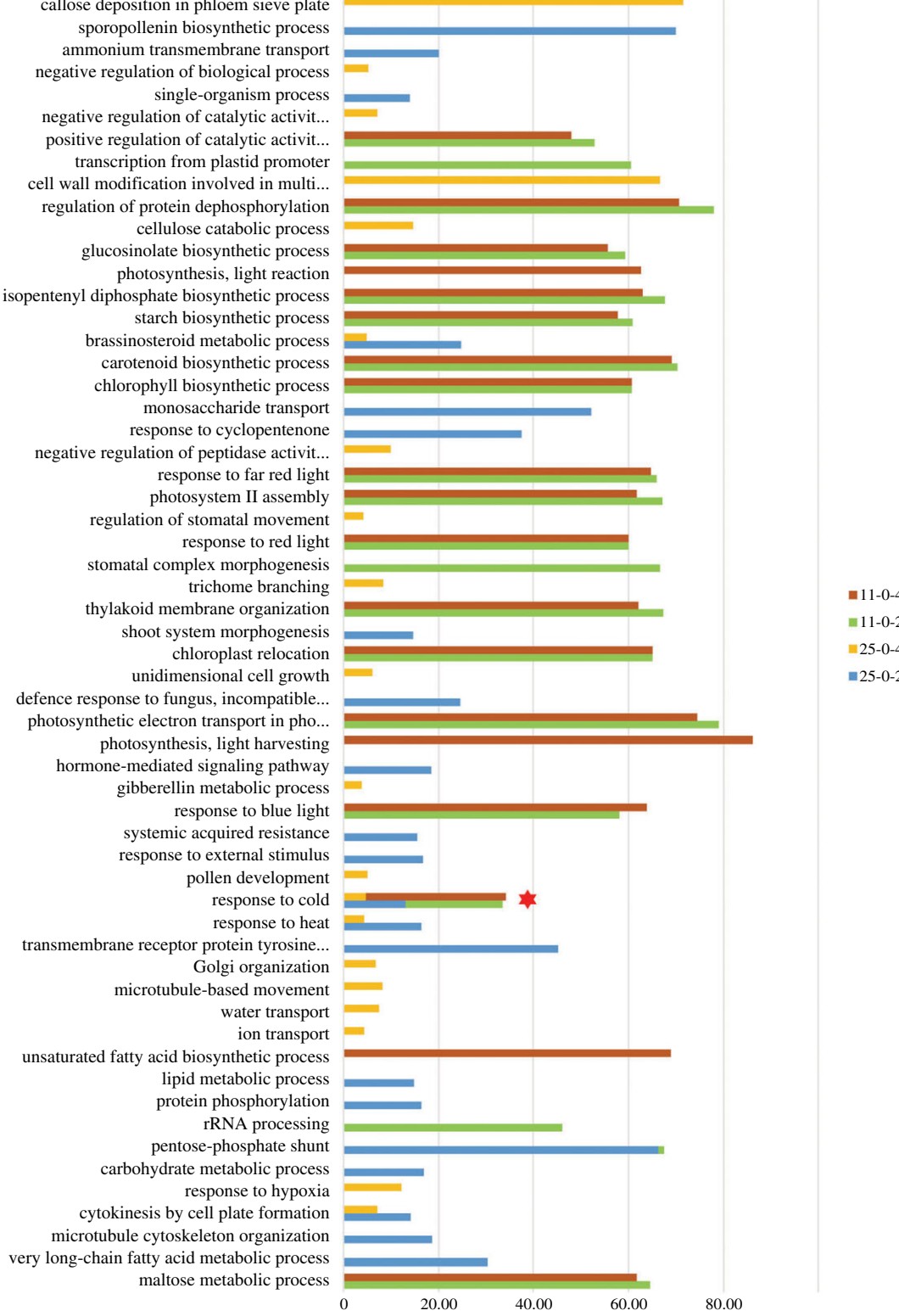

**Figure 4.** Top 20 biological progresses (BPs) GO terms with the highest percentage of significant genes of each compared group in LJ25 and LJ11 under 0, 2 and 4 days of chilling stress. Bars show the percentage of significant genes in each GO terms. The red star indicates that the GO term 'response to cold' was including in all of the compared groups.

supplementary material, table S6-1). With regard to hormone signalling, only three genes related to ABA were found. With regard to 'misc' functions, 11 genes were found related to beta-glucanase, two genes related to cell wall synthesis and one gene related to proteolysis. With regard to defence genes, two genes were detected; one gene was to secondary metabolite related, the other gene (*Os12g0244100*) was heat shock protein-related. Both

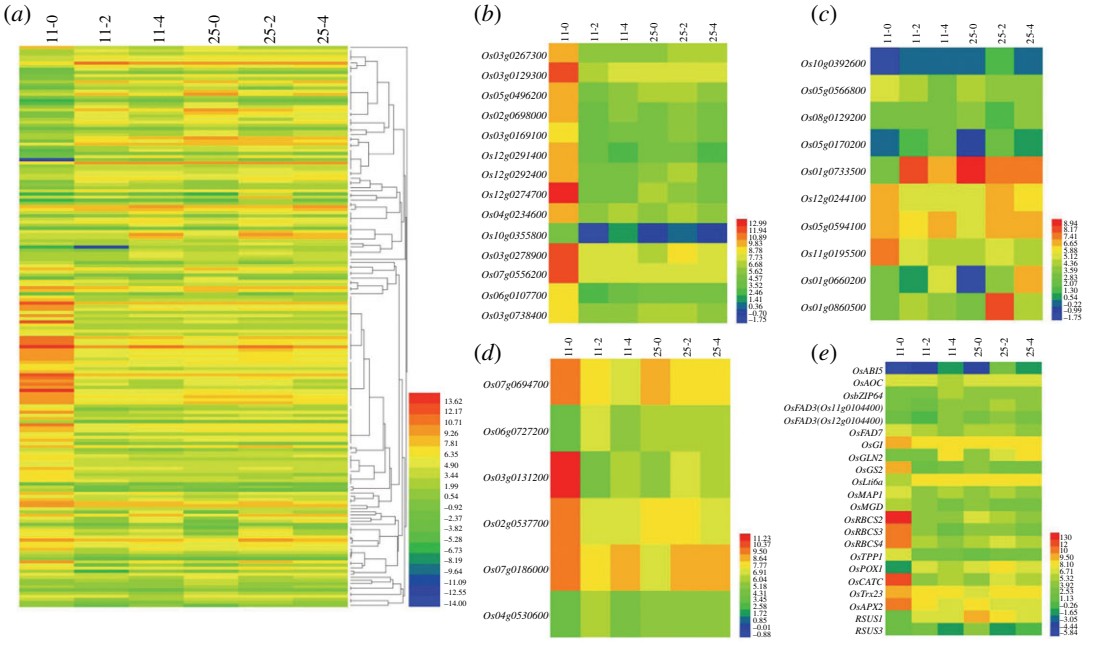

**Figure 5.** Heatmap displaying the change of relative expression of DEGs in LJ25 and LJ11 regarding GO term 'response to cold' during the entire period of chilling stress. (*a*) The dynamic changes of DEGs related to GO term 'response to cold' in chilling-tolerant and chilling-sensitive cultivars (LJ25 and LJ11) after 0, 2 and 4 days of chilling stress. (*b–e*) The dynamic changes of DEGs in LJ25 and LJ11 after 0, 2 and 4 days of chilling stress related to photosynthesis (*b*), abiotic stress and peroxidases (*c*), redox (*d*) and known genes (*e*), respectively.

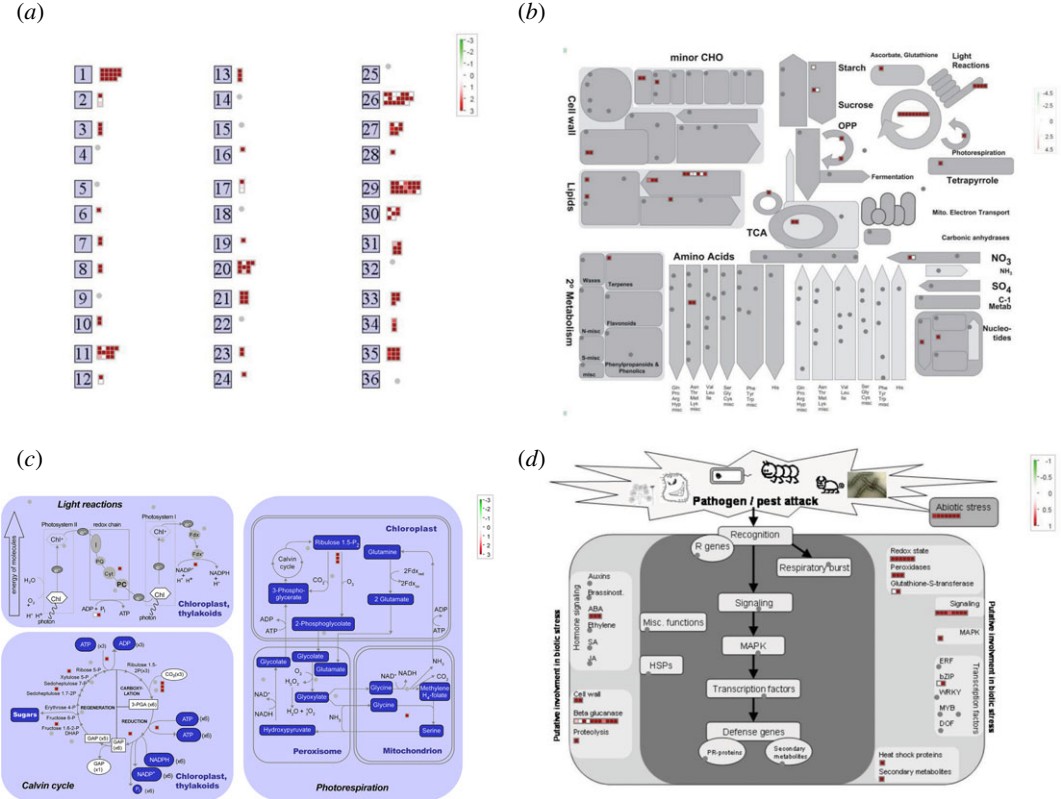

**Figure 6.** MAPMAN analysis of the DEGs involved in the GO term 'response to cold' in LJ25 and LJ11 after the entire chilling stress period. The total overview (*a*) and the overviews of metabolism (*b*), photosynthesis (*c*), and biotic stress (*d*) were mapped with the DEGs involved in the GO term 'response to cold'. Red boxes represent groups of genes upregulated by chilling stress, white boxes represent groups of genes whose expression was unaffected by chilling stress. Details are presented in electronic supplementary material, table S6.

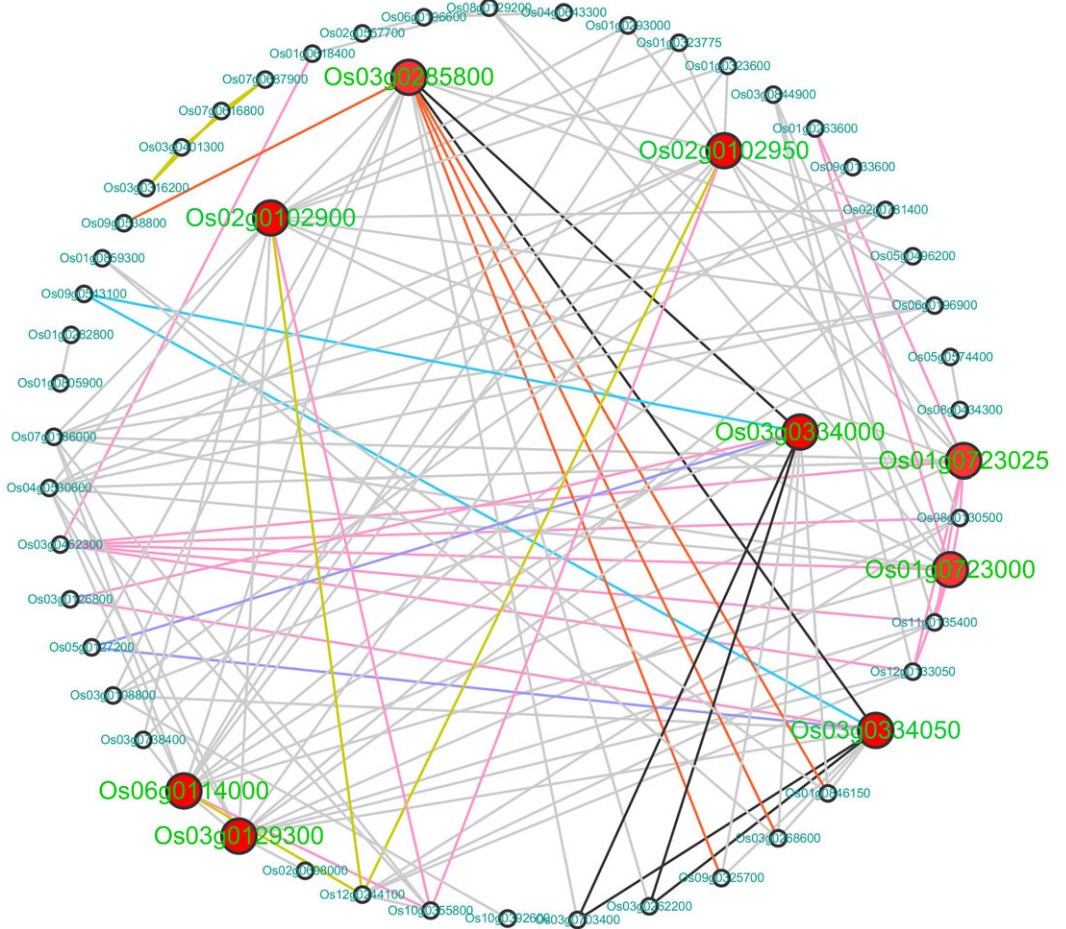

**Figure 7.** Interacting networks of DEGs regarding the GO term 'response to cold' in LJ25 and LJ11 following different chilling stress periods predicted by STRING and visualized in Cytoscape. The larger, red circles represent the hub genes with the gene IDs shown in green. The small, black circles represent other genes involved in interacting networks with the gene IDs shown in blue. Dark blue lines represent activation, light blue lines represent expression, and grey lines represent binding. Yellow lines represent catalysis, black lines represent inhibition, orange lines represent post-translational modification (ptmod), and pink lines represent reaction.

genes were downregulated in LJ11 and unaffected in LJ25 after chilling stress (figure 5*c*; electronic supplementary material, table S6–3). With regard to signalling, eight genes were detected, including one related to MAPK. Two bZIP-related genes were founded in the 'transcription factors' category, while in the 'MAPK category', only one gene (*Os03g0285800*) was found, known as *OsMAP1*. Seven genes were found to be related to abiotic stress, including three (*Os05g0566800*, *Os08g0129200*, *Os05g0170200*) that were cold stress-related, two (*Os01g0733500* and *Os05g0594100*) that were drought/salt and touch/wounding stress-related, respectively, one (*Os12g0244100*) that was heat stress-related and one (*Os10g0392600*) that was abiotic stress-related. Finally, with regard to respiratory burst, two genes were related to glutathione-S-transferases, while three genes were related to peroxidases. The abundant expression of these genes is shown in figure 5*c*. Six genes were related to redox reactions, of which two genes (*Os04g0530600* and *Os07g0186000*) were related to thioredoxin. One gene (*Os07g0694700*), known as *OsAPX2*, encoded L-ascorbate peroxidase 2. Two genes were related to dismutases and catalases. One gene, *Os06g0727200*, was upregulated in LJ11 and constitutively expressed in LJ25 under chilling stress, the other (*Os03g0131200*), was known as *OsCATC*. *Os02g0537700*, related to peroxiredoxin BAS1, showed the same expression pattern as *OsAPX2* and *OsCATC* (figure 5*d,e*; electronic supplementary material, table S6–4).

## 3.7. Functional network analysis of the DEGs in the GO category: 'response to cold'

To reveal the putative protein association network under chilling stress, 180 DEGs related to 'response to cold' were uploaded to the STRING database to predict their PPI network and visualized in Cytoscape. As showed in figure 7, 53 out of 180 DEGs interacted with each other. Among them, there were seven

interaction modes, and most were categorized to 'binding mode'. Among the 53 DEGs, 18 were protein-related, five were Calvin cycle-related, three each were cell-, lipid metabolism-, RNA-synthesis and abiotic stress-related, and a smaller number related to other categories (electronic supplementary material, table S6–6). Here we identified nine hub genes with the threshold of co-expressed gene number ≥11 (table 2).

## 3.8. Analysis of the antioxidant enzyme activities and the MDA content of the two rice varieties, LJ25 and LJ11, in response to chilling stress at the booting stage

We examined the antioxidant enzyme activities (POD, SOD, CAT) and the content of MDA of the two varieties after 0, 2 and 4 days of chilling stress (12°C), respectively, to determine their chilling tolerance (figure 8a–d). The activities of CAT and POD show similar trends in response to the chilling stress: both decreased markedly in LJ11, while both increased significantly in LJ25 during the first 2 days of chilling treatment and then remain constant. Moreover, the CAT activity in LJ11 after 4 days of chilling stress decreased to about 15% compared with its activity at day 0 (figure 8a,b). LJ11 exhibited a remarkable increase in SOD activity after 2 days of chilling treatment compared with the control, and then slightly decreased after 4 days of chilling treatment (although still higher than the control), while in LJ25, there were no changes in SOD activity during the entire period of chilling treatment (figure 8c). LJ11 exhibited a significantly higher MDA concentration compared with the control, especially after 4 days of chilling treatment. However, MDA decreased significantly in LJ25 after 2 days of chilling treatment, but then returned to normal after 4 days of chilling treatment (figure 8d). Taken together, these results indicated that LJ25 is more tolerant to chilling stress during the booting stage than LJ11.

## 3.9. Quantitative real-time PCR analysis to validate gene expression levels found by RNA-seq

To validate the gene expression level obtained by RNA-seq, 10 DEGs were randomly selected to perform qRT-PCR analysis (electronic supplementary material, table S1). As shown in figure 9, the expression profiles of all ten genes confirmed the original transcriptome data obtained by RNA-seq, suggesting that the RNA-seq analysis was reliable and that the transcriptome data obtained for the DEGs analysis was credible.

# 4. Discussion

Low temperature is a major restrictive factor for normal rice growth, especially in areas of high latitude and altitude. Some progress has been made in the breeding of chilling-tolerant rice. However, the molecular and genetic mechanisms of the response to chilling stress are complex and still need to be explored further [40,41]. In our study, we used RNA-Seq to evaluate the transcriptomes of the young spikelets of two different rice genotypes at the fertility-sensitive stage following chilling stress treatment. The data indicated that LJ11 is more sensitive to and displayed a fiercer response to chilling stress than LJ25. However, although the two cultivars are very similar in growth and plant type in field and are similar in many other agronomic traits except chilling tolerance at the booting stage, their genetic background has not been studied specifically. Therefore, we plan to carry out DNA re-sequencing on the LJ25 and LJ11 genomes to investigate their genetic differences in detail. Combined with the results of the transcriptome analysis, much more detail on the molecular and genetic mechanisms of the response to chilling stress in cold regions could be identified. Furthermore, in this study, GO function analysis revealed that the GO term 'response to cold' was significantly annotated in all groups that were compared.

## 4.1. Antioxidant system involved in chilling response during the booting stage of rice

ROS, such as superoxide ($O_2-$), hydrogen peroxide ($H_2O_2$) and the hydroxyl radical (HO−), are produced in high amounts under both biotic and abiotic stresses including chilling stress and can cause cellular oxidative damage [42]. Male sterility seems to be closely related to oxidative damage during pollen development. An increase in ROS and MDA and a decrease in POD, SOD and CAT activities occur in spikelets at the booting stage during chilling treatment [26]. As evident from our results, the activities of these enzymes were all significantly decreased in LJ11 under chilling stress,

**Table 2.** The annotation of the hub genes with co-expressed genes ≥11 identified in PPI analysis.

| Gene ID | PPI number | 11-0-2 FDR | 11-0-2 log$_2$FC | 11-0-4 FDR | 11-0-4 log$_2$FC | 25-0-2 FDR | 25-0-2 log$_2$FC | 25-0-4 FDR | 25-0-4 log$_2$FC | annotation |
|---|---|---|---|---|---|---|---|---|---|---|
| Os06g0114000 | 14 | $3.58 \times 10^{-23}$ | −2.70 | $1.01 \times 10^{-10}$ | −1.77 | $1.68 \times 10^{-2}$ | 0.85 | $6.92 \times 10^{-1}$ | 0.24 | TCP-1/cpn60 chaperonin family |
| Os02g0102900 | 14 | $9.78 \times 10^{-6}$ | −1.43 | $3.02 \times 10^{-6}$ | −1.46 | $2.67 \times 10^{-7}$ | −1.92 | $1.86 \times 10^{-2}$ | −1.09 | TCP-1/cpn60 chaperonin family |
| Os03g0285800 | 14 | $1.37 \times 10^{-12}$ | −3.42 | $1.34 \times 10^{-9}$ | −2.75 | $5.18 \times 10^{-10}$ | 1.90 | $9.27 \times 10^{-1}$ | 0.08 | protein tyrosine kinase |
| Os02g0102950 | 14 | $3.88 \times 10^{-2}$ | −1.38 | $7.47 \times 10^{-3}$ | −1.77 | $3.81 \times 10^{-3}$ | −2.12 | $1.30 \times 10^{-1}$ | −1.22 | RuBisCO large subunit-binding protein subunit beta |
| Os03g0129300 | 12 | $5.20 \times 10^{-30}$ | −4.61 | $6.88 \times 10^{-22}$ | −4.15 | $9.50 \times 10^{-1}$ | 0.03 | $7.49 \times 10^{-1}$ | −0.22 | glyceraldehyde 3-phosphate dehydrogenase, C-terminal domain |
| Os03g0334000 | 11 | $1.05 \times 10^{-14}$ | 2.43 | $9.65 \times 10^{-19}$ | 2.60 | $1.28 \times 10^{-2}$ | 0.85 | $4.65 \times 10^{-1}$ | 0.40 | protein tyrosine kinase |
| Os01g0723000 | 11 | $4.86 \times 10^{-18}$ | Inf | — | — | $1.88 \times 10^{-1}$ | −2.83 | $4.28 \times 10^{-1}$ | −1.37 | elongation factor Tu GTP binding domain |
| Os01g0723025 | 11 | $4.43 \times 10^{-86}$ | 7.51 | $8.80 \times 10^{-16}$ | 4.53 | $3.13 \times 10^{-1}$ | −1.95 | $5.01 \times 10^{-1}$ | −1.09 | elongation factor Tu domain 2 |
| Os03g0334050 | 11 | $1.28 \times 10^{-17}$ | 2.42 | $2.08 \times 10^{-15}$ | 2.23 | $3.10 \times 10^{-2}$ | 0.71 | $5.27 \times 10^{-1}$ | 0.35 | protein tyrosine kinase |

Note: '—' indicates low expressive level of genes; 'Inf$^p$' indicates extremely high expressive level.

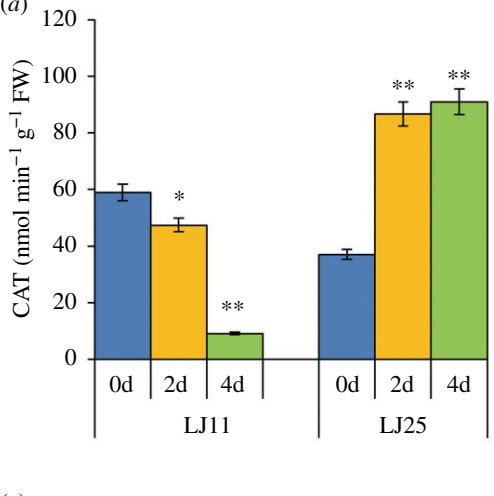

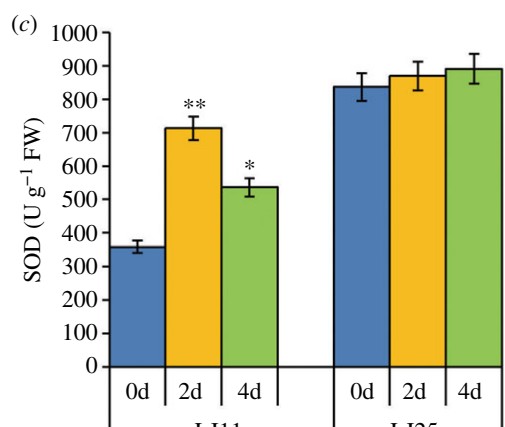

**Figure 8.** Effect of chilling stress on the activities of oxidants and antioxidant enzymes (CAT, POD and SOD) and on the contents of MDA in chilling-sensitive and chilling-tolerant rice plants. (*a*) Activity of CAT of LJ25 and LJ11 under different chilling stress periods. (*b*) Activity of POD of LJ25 and LJ11 under different chilling stress periods. (*c*) Activity of SOD of LJ25 and LJ11 under different chilling stress periods. (*d*) Content of MDA of LJ25 and LJ11 under different chilling stress periods. Data are mean ± s.e. of three replicates. Asterisks indicate statistically significant differences (*$p < 0.05$; **$p < 0.01$) between control group (0d) and treatment groups (2d and 4d). 0d: control (no chilling stress), 2d: 2 days of chilling stress; 4d: 4 days of chilling stress.

whereas either no change or a significant increase occurred in LJ25 (figure 5*e*). *OsCATC* [43,44], encoding catalase, was significantly downregulated in LJ11 compared with LJ25 under chilling stress. *OsPOX1* is a third peroxidase gene preferentially expressed in spikelets. It is expressed in vascular tissues and anthers at the microspore development stage, and is induced by low temperatures [45,46]. Indeed, *OsPOX1* was upregulated in both LJ11 and LJ25 (figure 5*e*), indicating that this gene may be constitutively expressed in rice in cold regions. Overexpression of *OsAPX2*, a cytoplasmic APX gene essential for rice pollen development, has been found to reduce the accumulation of ROS and MDA and enhance the tolerance to drought, salt and low temperature [47]. However, *OsAPX2* knockout mutants are male-sterile even in the absence of stress [48]. *OsAPX1* [26,49], another cytoplasmic APX gene, shows similar effects as *OsAPX2*. Our results show that *OsAPX2* is downregulated markedly in LJ11 compared with LJ25 under chilling stress (figure 5*e*).

## 4.2. MAPKK-MAPK signal transduction networks involved in chilling response during the booting stage of rice

MAPKK-MAPK signal transduction networks have been extensively explored in plants. *OsMAP1* [50], *OsMAPK5* [51], *OsMPK5* [52], *OsMAPK2* [53], *OsMAPK3* [45] and *OsMPK3* [54], are reported to lie on the same gene locus, and are induced by several kinds of biotic and abiotic stresses, including ABA, pathogen infection, mechanical wounding, drought, salt and low temperature. *OsMAP1* and *OsMEK1* are specially induced by incubation at 12°C for 4 days and possibly interact to involve the MAPK

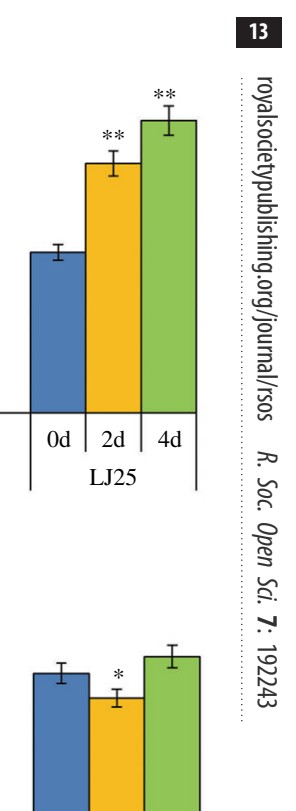

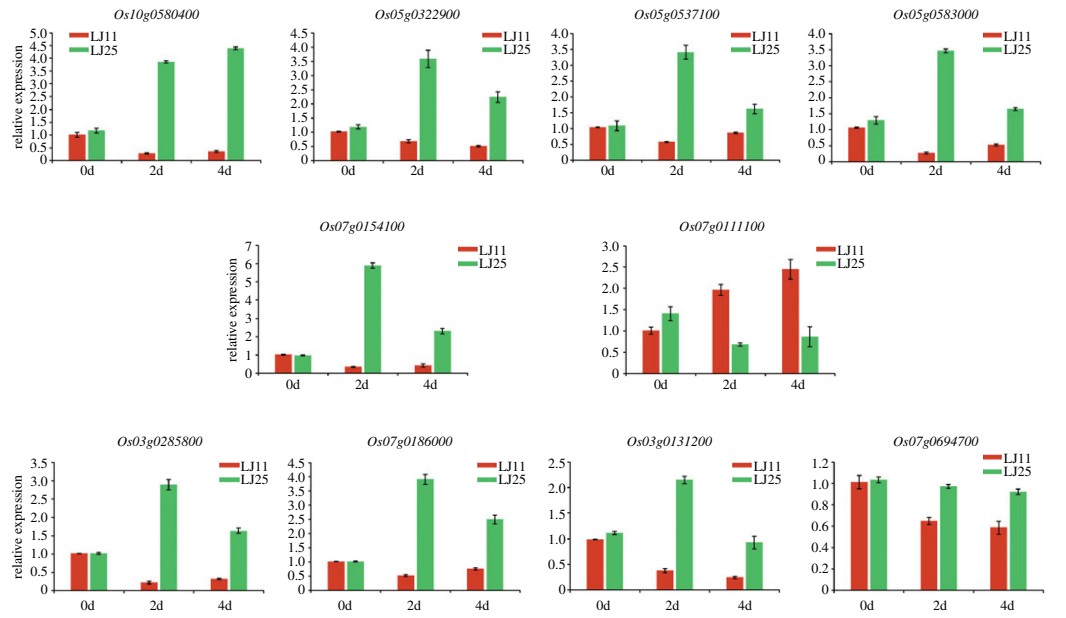

**Figure 9.** Quantitative real-time PCR (qRT-PCR) validated the relative expression levels of ten DEGs related to 0 day, 2 days and 4 days of chilling stress treatment at transcriptional level in LJ11 and LJ25. Values show the means (±s.d.) of three biological replicates.

signalling pathway for adaption to chilling stress [50]. Furthermore, *OsMAPK5* positively regulates the tolerance of rice to drought, salt and chilling stress, while it negatively regulates the expression of pathogenesis-related (*PR*) genes such as *PR1* and *PR10* [51]. *OsMPK3* physically interacts with and phosphorylates *SUB1A* (a submergence tolerant allele gene) [54]. Moreover, the tolerant allele *SUB1A* binds to the *OsMPK3* promoter and regulates its expression positively during submergence stress signalling. In our study, *OsMAP1* was markedly downregulated in LJ11 after chilling stress, while it was upregulated in LJ25 after 2 days chilling stress and subsequently returned to the expression level nearly that of the control (figure 5*e*). Thioredoxins (Trxs) are a multigene family of proteins with 10 members of the h-type Trxs in rice, which play a critical role in redox balance regulation [22]. *OsTrx23*, a major and cold-induced h-type Trx of rice, has an inhibitory activity on stress-activated *OsMPK3* and *OsMPK6 in vitro* [55]. Furthermore, *OsTRXh1* (*OsTrx23*) knockdown rice produced more $H_2O_2$ than wild-type rice, whereas its overexpression in rice caused the production of less $H_2O_2$ than wild-type [22]. Our results show that the expression of *OsTrx23* is decreased in LJ11 with continuous chilling treatment, whereas it is increased in LJ25 upon chilling treatment (figure 5*e*). Thus, the results for all genes described above suggest that LJ25 is more tolerant to chilling stress than LJ11, and these genes should be treated as candidate genes for the promotion of chilling tolerance in rice in cold regions.

## 4.3. The hub genes involved in the PPI network are crucial for chilling response during the booting stage of rice

As described in the PPI network, nine hub genes were found. Three of these genes (*Os06g0114000*, *Os02g0102900* and *Os02g0102950*), encoding subunits of the RuBisCO large subunit-binding protein, were co-expressed with 14 genes, respectively. RuBisCO is an important carboxylase in the C3 carbon reaction of photosynthesis and an indispensable oxygenase in photorespiration [56]. In addition, *OsRBCS2*, *OsRBCS3* and *OsRBCS4*, as known genes encoding RuBisCO, were also detected in our research. The expression of these genes markedly decreased in LJ11 but did not change much in LJ25 (figure 5*e*). *Os03g0129300* was co-expressed with 12 genes, encoding glyceraldehyde 3-phosphate dehydrogenase, an enzyme that catalyses the phosphorylation and oxidation of glyceraldehyde 3-phosphate in the presence of NAD+ and phosphoric acid to form 1, 3-diphosphoglycerate. However, these genes are all functionally related to photosynthesis, which mainly proceeds in leaves and rarely occurs in young panicles. Thus, we speculate these findings might reflect the expression patterns of genes in leaves, and this should be verified in leaves under chilling stress at a later date. Another three genes (*Os02g0102950*, *Os02g0102950* and *Os02g0102950*), encoding protein kinase domains, were co-expressed with 14, 11 and 11 genes, respectively. Two genes, (*Os01g0723000* and *Os01g0723025*), encoding the GTP binding domain of

elongation factor Tu, were both co-expressed with 11 genes, respectively. Moreover, these genes did not noticeably change their expression pattern in LJ25, whereas they were either down- or upregulated in LJ11 under chilling stress (table 2), indicating that this degree of chilling stress could not affect their normal expression in LJ25 but heavily disturbed their expression in LJ11. The strong co-expression of genes involved in PPIs revealed strong interaction among proteins involved in TCP regulation, protein synthesis and post-translational modification and implied that the nine hub genes will play important roles in the response to chilling stress.

# 5. Conclusion

In conclusion, the purpose of our study was to identify low temperature responsive genes that can be effectively used to increase the chilling tolerance of rice grown in cold regions. For this, we carried out comparative transcriptome analysis between the LJ25 (chilling-tolerant) and LJ11 (chilling-sensitive) varieties under chilling stress. Bioinformatics analyses using GO and KEGG enrichment showed that the DEGs revealed important biological processes and related metabolic and regulatory pathways, especially the GO term 'response to cold'. Furthermore, by applying MAPMAN tools and PPI network analysis of the DEGs in this term, we identified several chilling stress responsible genes, such as *OsCATC*, *OsTrx23*, *OsPOX1* and *OsMAP1*, which could be used as the candidate genes for molecular breeding programmes of rice in cold regions.

Data accessibility. All data generated or analysed during this study are included in this published article. Raw sequence data was deposited in the Sequence Read Archive (SRA) database (www.ncbi.nlm.nih.gov/sra), BioProject accession: PRJNA495106 (www.ncbi.nlm.nih.gov/bioproject).
Authors' contributions. Z.G. and L.C. conceived the original screening and research plans. G.P. and T.G. supervised the experiments. Z.G., C.H. and C.L. performed most of the experiments. Z.G. and L.C. designed the experiments and analysed the data. Z.G. conceived the project and wrote the article. Z.G. and L.C. supervised and complemented the writing. All authors read and approved the final version of the paper.
Competing interests. The authors declare that they have no conflict of interest.
Funding. This study was supported by The National Key Research and Development Program of China (grant no. 2016YFD0101801), The National Key Research and Development Program of China (grant no. 2017YFD0100503) and The Modern Agro-industry Technology Research System of China (grant no. CARS-01-09). The funders had no role in study design, data collection and analysis, decision to publish or preparation of the manuscript.
Acknowledgements. We thank the reviewers and the editor for helpful comments on this manuscript.

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
