## [Reviewer comments · Royal Society Open Science]

Review History

RSOS-192243.R0 (Original submission)

Review form: Reviewer 1

Is the manuscript scientifically sound in its present form?

Yes

Are the interpretations and conclusions justified by the results?

Yes

Is the language acceptable?

No

Do you have any ethical concerns with this paper?

No

Have you any concerns about statistical analyses in this paper?

No

Recommendation?

Accept with minor revision (please list in comments)

Comments to the Author(s)

The aim of this work is to elucidate the molecular mechanisms underlying the response to chilling stress during the booting stage in Japonica rice cultivars. The authors' strategy in the present work technically sounds, RNA sequencing and bioinformatic analyses appear to be conducted properly, and the data obtained are presented nicely. I agree to the main conclusion that several chilling stress responsible genes, such as CATC, Trx23, POX1, and MAP1, can be useful for breeding of cold-tolerant rice cultivars. Also, I believe that the present work will be potentially of interest to many scientists in a range of research areas such as plant stress physiology, signaling, and molecular biology.

I have only a few minor comments to improve the manuscript.

1. L139. It is better to explain more about genetic relationship between LJ25 and LJ11. How was LJ25 produced from LJ11?
2. L175-179. Experimental procedures for enzyme activity assay and MDA determination should be explained in detail. It is not fair only to cite a paper provided by another research group.
3. There are many grammatical/typographical errors. The authors should have checked more carefully before submission.

Decision letter (RSOS-192243.R0)

22-Apr-2020

Dear Dr ZHENHUA,

The editors assigned to your paper ("Global analysis of differentially expressed genes between two Japonica rice varieties induced by low-temperature during the booting stage by RNA-Seq") have now received comments from reviewers. We would like you to revise your paper in accordance with the referee and Associate Editor suggestions which can be found below (not including confidential reports to the Editor). Please note this decision does not guarantee eventual acceptance.

The Editor asks that you carefully revise the paper in accordance with the recommendations below. They also ask that you seek advice from a language editing service, such as <https://royalsociety.org/journals/authors/benefits/language-editing/>, before resubmitting.

Please submit a copy of your revised paper before 15-May-2020. Please note that the revision deadline will expire at 00.00am on this date. If we do not hear from you within this time then it will be assumed that the paper has been withdrawn. In exceptional circumstances, extensions may be possible if agreed with the Editorial Office in advance. We do not allow multiple rounds of revision so we urge you to make every effort to fully address all of the comments at this stage. If deemed necessary by the Editors, your manuscript will be sent back to one or more of the original reviewers for assessment. If the original reviewers are not available, we may invite new reviewers.

To revise your manuscript, log into <http://mc.manuscriptcentral.com/rsos> and enter your Author Centre, where you will find your manuscript title listed under "Manuscripts with

Decisions." Under "Actions," click on "Create a Revision." Your manuscript number has been appended to denote a revision. Revise your manuscript and upload a new version through your Author Centre.

- Data accessibility

If you wish to submit your supporting data or code to Dryad (<http://datadryad.org/>), or modify your current submission to dryad, please use the following link:
<http://datadryad.org/submit?journalID=RSOS&manu=RSOS-192243>

- Competing interests

- Authors' contributions

- Acknowledgements

- Funding statement

Kind regards,

Andrew Dunn

on behalf of Professor Xiaodong Zhang (Associate Editor) and Catrin Pritchard (Subject Editor)

Associate Editor's comments (Professor Xiaodong Zhang):

Associate Editor: 1

Comments to the Author:

The work presents transcriptomic analysis of two rice varieties (one chilling-tolerant and chilling-sensitive) in order to identify global changes in gene expression, thus genetic response that leads to chilling-tolerance. It provides useful new information that could lead to improved rice varieties. However, the manuscript needs to be improved before acceptance. Specifically

1) more and proper references need to be provided

2) more detailed methods need to be provided

3) Spelling and grammatical errors and language/style overall needs to be improved

4) Figure legends need more detailed description

5) In addition to show transcriptome differences, the authors should analyse/discuss differences between genetic differences between the two strains and relate the differences to the differences in transcriptome profiling

Comments to Author:

Reviewers' Comments to Author:

Reviewer: 1

Comments to the Author(s)

The aim of this work is to elucidate the molecular mechanisms underlying the response to chilling stress during the booting stage in Japonica rice cultivars. The authors' strategy in the present work technically sounds, RNA sequencing and bioinformatic analyses appear to be conducted properly, and the data obtained are presented nicely. I agree to the main conclusion that several chilling stress responsible genes, such as CATC, Trx23, POX1, and MAP1, can be useful for breeding of cold-tolerant rice cultivars. Also, I believe that the present work will be potentially of interest to many scientists in a range of research areas such as plant stress physiology, signaling, and molecular biology.

I have only a few minor comments to improve the manuscript.

1. L139. It is better to explain more about genetic relationship between LJ25 and LJ11. How was LJ25 produced from LJ11?

2. L175-179. Experimental procedures for enzyme activity assay and MDA determination should be explained in detail. It is not fair only to cite a paper provided by another research group.
3. There are many grammatical/typographical errors. The authors should have checked more carefully before submission.

Author's Response to Decision Letter for (RSOS-192243.R0)

See Appendix A.

Decision letter (RSOS-192243.R1)

01-Jun-2020

Dear Dr ZHENHUA,

It is a pleasure to accept your manuscript entitled "Global analysis of differentially expressed genes between two Japonica rice varieties induced by low-temperature during the booting stage by RNA-Seq" in its current form for publication in Royal Society Open Science.

on behalf of Professor Xiaodong Zhang (Associate Editor) and Catrin Pritchard (Subject Editor)
openscience@royalsociety.org

Appendix A

Dear editor:

Thank you for arranging a timely review of our manuscript (RSOS-192243.R1), entitled *Global analysis of differentially expressed genes between two Japonica rice varieties induced by low-temperature during the booting stage by RNA-Seq*. We are pleased to know that our study received highly favorable comments. We have carefully evaluated the reviewers' comments and thoughtful suggestions, responded to these suggestions point-by-point, and revised the manuscript accordingly. Here, we attached a revised manuscript in Word format with tracked changes for your approval.

We sincerely hope our revised manuscript meets the standard of Royal Society Open Science and will be accepted for publication.

Yours sincerely

Zhenhua Guo

Associate Editor's comments (Professor Xiaodong Zhang):

Associate Editor: 1

Comments to the Author:

The work presents transcriptomic analysis of two rice varieties (one chilling-tolerate and chilling-sensitive) in order to identify global changes in gene expression, thus genetic response that leads to chilling-tolerance. It provides useful new information that could lead to improved rice varieties. However, the manuscript needs to be improved before acceptance. Specifically

1) more and proper references need to be provided

Response: Thank you for your meticulous and considerate comment. We have added additional, appropriate references containing the methods to measure the activities of SOD, POD, CAT and the MDA content and deleted the less relevant references in our revised manuscript.

2) more detailed methods need to be provided

Response: Thank you for this comment. We have described the methods used to measure the activities of SOD, POD, CAT and the MDA content in more detail in our revised manuscript.

3) Spelling and grammatical errors and language/style overall needs to be improved

Response: Thank you for your comment. We have carefully examined the spelling and

grammatical errors and the improper descriptions throughout the manuscript carefully and, with the help of the 'MJ Language Editing Services', we have corrected the spelling and grammatical errors in our revised manuscript, and the editorial certificate was attached below.

4) Figure legends need more detailed description

Response: Thanks very much for your comment. According to your advice, we have added more details in the figure legends of Fig.1, Fig.2, Fig.4, Fig.5, Fig.6, Fig.7, Fig.8, and Fig.9 in our revised manuscript.

5) In addition to show transcriptome differences, the authors should analyse/discuss differences between genetic differences between the two strains and relate the differences to the differences in transcriptome profiling

Response: Thank you very much for this comment. According to your advice, we have discussed the genetic differences between LJ25 and LJ11 and analyzed the relationship of the genetic differences and the transcriptome differences between LJ25 and LJ11 in the discussion part of our revised manuscript.

Comments to Author:

Reviewers' Comments to Author:

Reviewer: 1

Comments to the Author(s)

The aim of this work is to elucidate the molecular mechanisms underlying the response to chilling stress during the booting stage in Japonica rice cultivars. The authors' strategy in the present work technically sounds, RNA sequencing and bioinformatic analyses appear to be conducted properly, and the data obtained are presented nicely. I agree to the main conclusion that several chilling stress responsible genes, such as CATC, Trx23, POX1, and MAP1, can be useful for breeding of cold-tolerant rice cultivars. Also, I believe that the present work will be potentially of interest to many scientists in a range of research areas such as plant stress physiology, signaling, and molecular biology.

I have only a few minor comments to improve the manuscript.

1. L139. It is better to explain more about genetic relationship between LJ25 and LJ11. How was LJ25 produced from LJ11?

Response: Thank you very much for this rigorous and meticulous comment. We regret the confusion due to the description of the mutant and wild type for LJ25 and LJ11. In fact, the two cultivars LJ25 and LJ11 do not have a relationship of mutant and wild type. Since LJ25 was derived from a cross of Jiahezaozhan and Longhua97058 (<http://www.ricedata.cn/variety/varis/605744.htm>), and LJ11 was derived from a cross of Sha29 and Hejiang21 (<http://www.ricedata.cn/variety/varis/605957.htm>). However, most of the parental strains of LJ25 and LJ11 were generated from the backbone parents for rice breeding in Heilongjiang Province. Therefore, the two cultivars have a similar genetic background and share a close genetic relationship. Moreover, they are very similar in growth and plant type in the field and in many other agronomic traits. In spite of the similarities in those agronomic traits, there is a significant difference in the chilling tolerance at the booting stage between these two cultivars, since LJ25 has acquired chilling tolerance while LJ11 has remained chilling sensitive. Based on this, LJ25 and LJ11 are widely used for chilling tolerance analysis, and treated as “mutant” and “wild type”, respectively, though they are not technically in that relationship. Again, we apologize for this inaccurate statement and we have made appropriate changes in our revised manuscript.

2. L175-179. Experimental procedures for enzyme activity assay and MDA determination should be explained in detail. It is not fair only to cite a paper provided by another research group.

Response: Thanks for reviewer’s reminding. I realize that it was really too rough to explain the experimental procedures for enzyme activity assay and MDA determination. Now I’ve rewritten this part and given more details of experimental procedures in the revised manuscript.

3. There are many grammatical/typographical errors. The authors should have checked more carefully before submission.

Response: Thank you very much for this comment. We have checked the grammatical/typographical errors carefully throughout the manuscript and corrected them in our revised manuscript.

EDITORIAL CERTIFICATE

The English writing of the following manuscript was carefully edited by a native English speaker.

Manuscript information

ID: MJ202005142652

Editing date: 2020.05.14

Title: Global analysis of differentially expressed genes between two Japonica rice varieties induced by low-temperature during the booting stage by RNA-Seq

Author: Zhenhua Guo, Lijun Cai, Chuanxue Liu, Cuihong Huang, Zhiqiang Chen, Guojun Pan, Tao Guo

Language writing

before editing Very poor Poor Fair Good Very good Excellent

Recommendation Submitting to target journal directly

after language

editing Submitting to target journal after minor revision

Re-editing required after major revision

Not suitable for publication

Certificate by

Saphiya. K

Editor in Chief

MJ Language Editing Services, Shenzhen, China

MJ Language Editing Services, offers professional English language editing and publication support services to authors engaged in over 500 areas of research through its community of experienced editors, which includes doctors, published scientists, and researchers with peer review experience. Authors who work with MJ are guaranteed excellent language quality and timely delivery.

MJ Language Editing Services

Diwang Building, No. 5002 Shennan Road, Luohu District, Shenzhen, China

Tel: +086 0755 25100506